# Evaluation of the Universal Salt Iodization (USI) surveillance system in Tanzania, 2022

David Mahwera[1,2], Erick Killel[3], Ninael Jonas[1,2], Adam Hancy[4], Anna Zangira[3], Aika Lekey[3], Rose Msaki[3], Doris Katana[5], Rogath Kishimba[2,6], Debora Charwe[4], Fatma Abdallah[3], Geofrey Chiduo[4], Ray Masumo[3], Germana Leyna[1,3], Geofrey Mchau[1,2,3]*

1 Department of Epidemiology and Biostatistics, Muhimbili University of Health and Allied Sciences, Dar es salaam, Tanzania, 2 Tanzania Field Epidemiology and Laboratory Training Program, Tanzania, 3 Department of Community Health and Nutrition, Tanzania Food and Nutrition Centre, Dar es salaam, Tanzania, 4 Department of Nutrition Policy and Planning, Tanzania Food and Nutrition Center, Dar es salaam, Tanzania, 5 Department of Nutrition Education and Training, Tanzania Food and Nutrition Centre, Dar es salaam, Tanzania, 6 Ministry of Health, Dodoma, Tanzania

* gmchau80@gmail.com

## Abstract

### Background

The evaluation of surveillance systems has been recommended by the World Health Organization (WHO) to identify the performance and areas for improvement. Universal salt iodization (USI) as one of the surveillance systems in Tanzania needs periodic evaluation for its optimal function. This study aimed at evaluating the universal salt iodization (USI) surveillance system in Tanzania from January to December 2021 to find out if the system meets its intended objectives by evaluating its attributes as this was the first evaluation of the USI surveillance system since its establishment in 2010. The USI surveillance system is key for monitoring the performance towards the attainment of universal salt iodization (90%).

### Methodology

This evaluation was guided by the Center for Disease Control Guidelines for Evaluating Public Health Surveillance Systems, (MMWR) to evaluate USI 2021 data. The study was conducted in Kigoma region in March 2022. Both Purposive and Convenient sampling was used to select the region, district, and ward for the study. The study involved reviewing documents used in the USI system and interviewing the key informants in the USI program. Data analysis was done by Microsoft Excel and presented in tables and graphs.

### Results

A total of 1715 salt samples were collected in the year 2021 with 279 (16%) of non-iodized salt identified. The majority of the system attributes 66.7% had a good performance with a score of three, 22.2% had a moderate performance with a score of two and one attribute with poor performance with a score of one. Data quality, completeness and sensitivity were 100%, acceptability 91.6%, simplicity 83% were able to collect data on a single sample in <

**Data Availability Statement:** All relevant data are within the manuscript and its Supporting information files.

**Funding:** The author(s) received no specific funding for this work.

**Competing interests:** The authors have declared that no competing interests exist.

2 minutes, the system stability in terms of performance was >75% and the usefulness of the system had poor performance.

## Conclusion

Although the system attributes were found to be working overall well, for proper surveillance of the USI system, the core attributes need to be strengthened. Key variables that measure the system performance must be included from the primary data source and well-integrated with the Local Government (district and regions) to Ministry of Health information systems.

## Introduction

Iodine deficiency disorders remain a significant global health concern, affecting nearly two billion individuals around the world and exposing them to the risk of enduring irreversible brain damage, stillbirth, and cognitive impairments [1–3]. Despite the ongoing challenge of iodine deficiency disorders, several studies have revealed that individuals who do not consume iodized salt remain vulnerable to developing these conditions [2]. In Tanzania, iodine deficiency and its associated disorders are acknowledged as a substantial public health concern of micronutrient deficiencies [4].

Universal Salt Iodization (USI) is a worldwide strategy that has demonstrated remarkable effectiveness in the prevention and control of Iodine Deficiency Disorders (IDD). The World Health Organization (WHO) recommends that to attain the USI objective, at least 90% of households should consume adequately iodized salt. Tanzania adopted the USI plan as a proactive approach to boost the consumption of adequately iodized salt and combat IDD issues since the 1990s. Multiple studies conducted in diverse locations indicate the efficacy of this strategy in mitigating iodine deficiency disorders (IDD) [3, 5]. It is imperative to maintain a systematic monitoring and evaluation of the USI system to ensure salt is iodized to the prescribed standards and that the public consistently consumes the recommended amount, thereby reducing the risk of iodine deficiency-related disorders [6]. Despite the global target, which necessitates 90% of households in the country to use iodized salt [7], only 61.2 percent of households in Tanzania were found to use iodized salt with an iodine content of 15 ppm. This discrepancy highlights the need for further examination and intervention to bridge the gap between the current utilization of iodized salt and the desired coverage for safeguarding public health [4].

The Universal Salt Iodization (USI) surveillance system is the framework established by the Tanzanian Food and Nutrition Centre (TFNC) for the comprehensive monitoring and assessment of salt iodization programs at all levels within Tanzania [8]. Tanzania has undertaken various initiatives to establish this system, notably the formation of the Tanzania Salt Producers Association (TASPA) in 1994, the establishment of the National Council for the Control of Iodine Deficiency Disorders (NCCIDD) in 1985, and the introduction of the Iodated Salt Regulation under The Tanzania Food, Drugs, and Cosmetics (TFDA) Act (CAP 219) in 1994/5, with a subsequent review in 2010. These regulations included penalties for non-compliance. Notably, the TFDA Act came into effect in the same year the surveillance system was established (2010) [8]. The TFDA Act encompasses several critical provisions, such as restrictions on the importation of non-iodized salt, mandatory requirements for edible salt, and a provision that mandates an authorized officer to inspect and analyze iodized salt. Following the production of inspection reports, there exists a structured reporting mechanism, ensuring a clear

path for information flow from lower levels to its final destination [11]. In 2019, the TFDA was restructured into the Tanzania Medicine and Medical Devices Authority (TMDA), while the Tanzania Bureau of Standards (TBS) assumed responsibility for all food-related matters, including the USI surveillance system.

A schematic pathway outlining the major milestones and achievements of Tanzania's Universal Salt Iodation (USI) program is pivotal in understanding the program's trajectory and impact. Since its inception in the late 1990s, the program has undergone significant legislative, infrastructural, and educational developments. Initially, legislative measures were enacted to mandate salt iodization, laying the foundation for subsequent policy development and standardization efforts. Infrastructure investments followed suit, bolstering salt production facilities and quality control mechanisms to ensure compliance with iodization standards. Concurrently, extensive public awareness campaigns were launched, aiming to educate communities about the importance of iodized salt and dispel misconceptions. As the program progressed, partnerships were forged with international organizations and the private sector, facilitating resource mobilization and program scalability. Through regular monitoring and surveillance, the program tracked iodized salt coverage and iodine levels, enabling evidence-based decision-making and targeted interventions. By 2021, positive indicators such as increased iodized salt coverage and improved iodine status among populations underscored the program's effectiveness in combating iodine deficiency disorders. However, to comprehensively evaluate its progress, there is a pressing need for detailed case studies examining the functioning and impact of the surveillance system. Such studies would provide invaluable insights into the program's strengths, weaknesses, and areas for refinement, ensuring its continued success in promoting public health and well-being across Tanzania.

Introducing a schematic pathway outlining the major timelines and achievements of the Universal Salt Iodation (USI) program in Tanzania is crucial for several reasons. Firstly, it provides a clear visual representation of the program's evolution, highlighting key milestones and initiatives undertaken since its inception. This pathway serves as a valuable resource for stakeholders, researchers, and policymakers to understand the trajectory of the USI program and its impact on addressing iodine deficiency disorders (IDDs) in Tanzania. Additionally, by tracing the program's progress over time, it enables us to assess the effectiveness of interventions, identify areas of success, and pinpoint challenges that may need further attention. However, while the pathway offers a broad overview, there remains a need for in-depth case studies to delve into specific aspects of the program's surveillance system and evaluation mechanisms. These case studies can provide valuable insights into how the surveillance system functions, its strengths and weaknesses, and the overall impact of the USI program on iodine status and public health outcomes. By conducting detailed evaluations and assessments, we can ensure that the USI program continues to evolve and adapt to meet the changing needs of the population, ultimately advancing the goal of sustainable salt iodization and improved health outcomes for all Tanzanians.

In India, a comprehensive study conducted by Research Institute for Compassionate Economics (RICE) focused on evaluating the effectiveness of salt iodization programs in rural areas. Methodology: The study employed a mixed-method approach, combining household surveys, salt sample testing, and qualitative interviews with stakeholders. This multifaceted approach allowed researchers to assess not only the coverage and quality of iodized salt but also the socio-economic factors influencing its consumption [9].

A study conducted in Ethiopia by the Ethiopian Public Health Institute evaluated the impact of salt iodization programs on iodine status among school-aged children. Methodology: This study utilized a longitudinal design, tracking changes in urinary iodine concentration

(UIC) among a cohort of schoolchildren over time. Additionally, dietary assessments and salt sample testing were conducted to correlate iodine intake with iodized salt consumption [10].

Researchers from the International Centre for Diarrhoeal Disease Research, Bangladesh (icddr,b) conducted a study to assess the sustainability of salt iodization programs in urban slum areas. Methodology: The study employed a community-based participatory research (CBPR) approach, involving collaboration with local communities to monitor salt iodization coverage and compliance. This participatory methodology empowered community members to take ownership of iodized salt consumption and advocate for its availability [11].

Our study offers a unique contribution to Universal Salt Iodization (USI) evaluation, guided by the Center for Disease Control Guidelines for Evaluating Public Health Surveillance Systems (MMWR) to assess USI 2021 data. Conducted in the Kigoma region in March 2022, our research provides a localized perspective, capturing unique challenges and successes within this specific context. Utilizing both Purposive and Convenient sampling techniques, our methodology ensures comprehensive representation across different levels of the USI program. Additionally, our incorporation of document review and key informant interviews enriches the depth and context of our analysis Overall, our study's adherence to guidelines, localized focus, and comprehensive methodology make it a valuable addition to the literature on USI evaluation, warranting publication consideration.

Tanzania's experience with universal salt iodization offers a compelling case study for the broader global context. Understanding the progress made in reducing goiter rates, monitoring urinary iodine excretion, and assessing the current status of iodine nutrition is essential for policymakers and public health experts alike. Evaluating how this surveillance works in Tanzania will help to examine the outcomes and challenges in Tanzania, which can provide valuable insights into the efficacy of universal salt iodization as a public health intervention and the necessity of sustaining a surveillance system to ensure optimal iodine intake for the population.

Therefore, the current study aimed to evaluate the universal salt iodization (USI) surveillance system in Tanzania from January to December 2021 to find out if the system meets its intended objectives. This evaluation utilized information collected by the Integrated Management and Evaluation System (IMES), a system which is part of District Health Information System (DHIS) II systems used by MOH and collects information from the district to national and Multisectoral Nutrition Information System (MNIS) managed by Tanzania Food and Nutrition Center (TFNC).

## Materials and methods

### Study design

This was a surveillance system evaluation. Public health surveillance entails the continuous, organized gathering, analysis, interpretation, and dissemination of information related to a health event. The primary goal is to inform public health interventions, mitigate morbidity and mortality, and enhance overall health outcomes. The objective of scrutinizing public health surveillance systems is to ensure the efficient and effective monitoring of significant public health issues. Regular evaluations of these systems should encompass recommendations aimed at enhancing quality, efficacy, and utility. This evaluation of the public surveillance system was guided by the Center for Disease Control Guidelines for Evaluating Public Health Surveillance Systems, (MMWR) to evaluate USI 2021 data.

### Study area

This surveillance system evaluation was conducted in the Kigoma region. The region was purposive selected based on the performance level. Within Kigoma region, Kigoma MC, Kasulu

TC, and Kibondo DC ware selected using both Purposive and Convenience sampling. These councils were purposively selected based on performance level informed by regional nutritionists, and conveniently selected based on the accessibility simply because they are along the road. Time limitation and budget insufficient was another factor that led to use convenience sampling to select Councils that was easily accessible during the time of evaluation. Within each district, Wards were conveniently selected based on the availability/presence of Ward health officers and then randomly selected to obtain the ward that participated in the study. Convenient was done in selecting Wards since not all Wards have health officers.

## Study population

All government officials who operate within the USI surveillance system play a crucial role in monitoring and overseeing salt iodization programs in Tanzania. This network of officials extends across various levels, from the national level, down to the regional, district, and ward levels. Their collective efforts are instrumental in ensuring that the objectives of the USI initiative are met effectively.

At the national level, policymakers and administrators work closely with the TFNC to formulate and implement policies that guide the USI program. This includes the establishment of regulatory frameworks and the development of strategies to promote the widespread use of adequately iodized salt. The TFNC, as a key body in this effort, collaborates with other agencies such as the Tanzania Medicine and Medical Devices Authority (TMDA) and the Tanzania Bureau of Standards (TBS) to enforce regulations and standards related to salt iodization.

Moving to the regional and district levels, government officials are responsible for overseeing the implementation of USI practices within their respective regions and districts. This entails working closely with salt producers, salt retailers, and local communities to ensure the availability and consumption of iodized salt. They also play a vital role in monitoring and reporting on compliance with salt iodization regulations, as set out in the Iodated Salt Regulation under The Tanzania Food, Drugs, and Cosmetics (TFDA) Act.

At the ward level, officials ensure that the USI program is effectively executed within their jurisdictions. They engage with local salt producers and traders, raise awareness among community members about the importance of iodized salt, and report any non-compliance with iodization standards to higher authorities. This multi-level structure ensures a well-coordinated effort to combat iodine deficiency disorders and maintain the quality and safety of iodized salt throughout the nation.

In summary, government officials at all levels, from the national to the ward level, are integral to the success of the USI surveillance system in Tanzania. The success of the USI surveillance system in Tanzania is dependent on them because they are the ones that engage in decision making, resource allocation and policy implementation as explained above. Government officials are responsible for implementing and enforcing policies related to salt iodization. Their commitment ensures that the regulations and guidelines are effectively carried out, contributing to the success of the USI surveillance system in ensuring that salt is adequately iodized to address public health concerns. e.g., the Tanzania Food and Drug Authority (TFDA) Act,2003 and the Salt Regulation Act,2010 recognize and task the health officers as one of government officials working in USI surveillance system to deal with the issue concerning salt iodization enforcement (TFDA Act section 105) [12–14].

## Data collection procedure and surveillance evaluation techniques

Documents and reports used in the USI surveillance system (Record Review) and Key informant interviews were used as the method of data collection for this evaluation. A total of 12

key informants were interviewed, with each interview questionnaire assigned a unique identification number for privacy. Data collection procedure was done according to all levels that are working on the USI surveillance system. The following are clear and direct ways in which data collection was conducted according to each level as also shown in Fig 1 below.

## National level data collection

The source of data at the national level was IMES system. Key informant interviews were conducted with selected staff who are responsible for USI surveillance system. Two staff members from the TFNC and the President's Office–Regional Administration and Local Government (PO-RALG) were chosen to provide data that are required.

## Regional, district (council) and ward level data collection

The source of data for this level was also IMES system. Key informant interview was also used as the method of data collection where regional health officer (RHO) and Regional Nutrition Officer (RNuO) were chosen as key informant using purposive sampling techniques as they are responsible for overseeing USI surveillance system at regional level. The permission to interview them was sought from Regional Medical Officer (RMO). For the Council level two interviewees were selected at each selected council (Kigoma Ujiji Council, Kasulu Urban, and Kibondo District). For this level, two Ward Health Officers were selected and permission was obtained from the council-level.

## Timeline of evaluation

The evaluation was conducted in March 2022. Data for the evaluation was extracted from the 2021 Universal Salt Iodization surveillance data at the national, regional, and council levels as explained in the data collection procedure.

At every level, from the national to the regional, district, and ward levels, an extensive data collection and assessment process was diligently conducted for each attribute under evaluation. These attributes, critical to the effectiveness of the surveillance system, included Usefulness, Flexibility, Stability, Representativeness, Data Quality, Acceptability, Timeliness, and Sensitivity.

To ensure a comprehensive evaluation, a range of carefully chosen indicators were scrutinized for each of these attributes, helping us determine if the surveillance system was effectively fulfilling its intended purpose. The apex of these assessments is presented in Tables 1 and 2 below, which provide a clear depiction of the performance of the surveillance system across various levels.

## Data processing and analysis

The evaluation process adhered to the guidelines provided by the Centers for Disease Control and Prevention (CDC), as outlined in the Morbidity and Mortality Weekly Report (MMWR) [15]. Subsequently, the data collected was entered and analyzed using Microsoft Excel. To assess the surveillance system's alignment with its intended objectives, we examined ten key attributes, namely simplicity, flexibility, data quality, acceptability, sensitivity, predictive value positive, representativeness, timeliness, stability, and usefulness. Each attribute was scrutinized for a set of specific indicators, with each indicator being assigned a score of one point.

The methodology employed for the evaluation entailed aggregating the scores for each attribute, followed by dividing the sum by the total number of indicators applied to assess that particular attribute. This calculation provided an average score for the attribute.

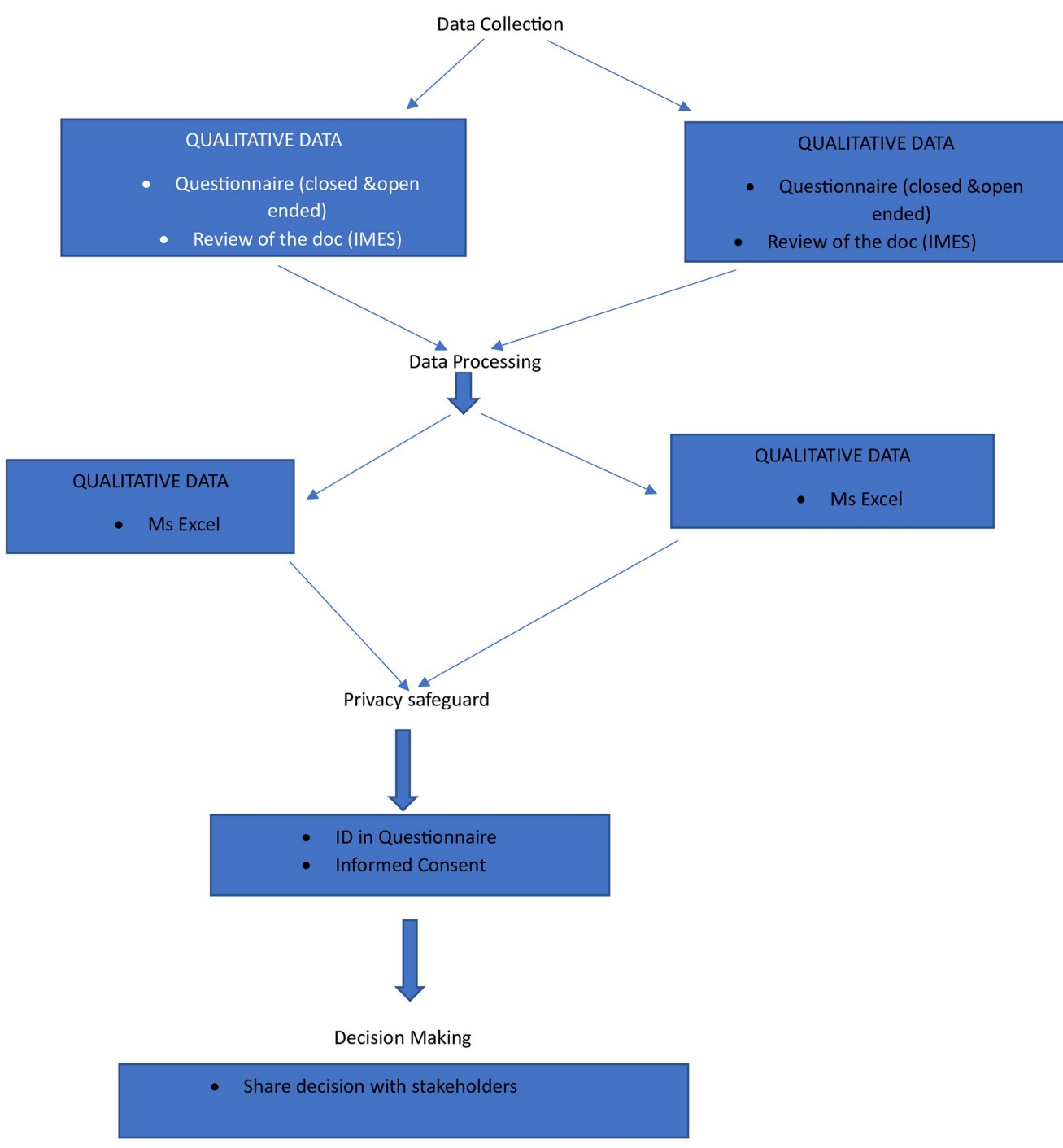

**Fig 1. Schematic design pathway of USI surveillance system evaluation.**

**Table 1. Evaluation of qualitative attribute indicators in the USI surveillance system, key findings, assessment scores (0 = key finding does not support the attribute, 1 = key finding supports the attribute), and overall scores (1 to 3)—Tanzania, 2021.**

| System Attributes | Indicator evaluated | Key finding | Assessment score | Overall score |
|---|---|---|---|---|
| Simplicity | • Amount and type of data necessary to establish that health-related event has occurred. | • A simple working definition | 1 | 3 |
| | • Time spent collecting, maintaining, analyzing, and disseminating USI data. | • Little time is taken to collect and analyze information on the single sample | 1 | |
| | • training is needed to conduct USI surveillance system | • Special training required | 0 | |
| Flexibility | • Adaptation to change (new case, new reporting sources, or change in technology | • Change of mandate from TFDA to TBS | 1 | 3 |
| | • Use of standard data format | • Follow the standard form of information and data in electronic form | 1 | |
| Stability | • The number of unscheduled outages and downtimes for the system's computer | • None | 1 | 3 |
| | • The percentage of time the system is operating fully | • >75% | 1 | |
| | • The sources of funds | • Government funded | 1 | |
| Representativeness | • Sample collected by place | • No description by place | 0 | 2 |
| | • Sample collected represents household samples | • Yes | 1 | |
| Usefulness | • Has the system met its objective? | • 1 out of 5 (20%) objectives were met | 0 | 1 |
| | • Number of iodized salt detection | • Non-iodized salt detected | 1 | |
| | • Generating reports for action taking | • No report generated | 0 | |
| | • Provide the magnitude of the problem | • The system has no target, hence difficult to give the magnitude of the problem | 0 | |

**Table 2. Evaluation of quantitative attribute indicators in the USI surveillance system, key findings, assessment scores (0 = key finding does not support the attribute, 1 = key finding supports the attribute), and overall scores (1 to 3)—Tanzania, 2021.**

| System Attributes | Indicator evaluated | Key findings | Assessment score | Overall score |
|---|---|---|---|---|
| Data quality 1. Data Completeness | • % Of missing record<br>N: Number of records in reporting form that are missing.<br>D: Total number of records | 900/900<br>(100%) | 1 | 3 |
| | • % Of incompletely filled forms in IMES.<br>N: Number of empty filled entry<br>D: Total number of entry fields | 60/60<br>(100%) | 1 | |
| 2. Data Accuracy | • A proportional mismatch between data in registers and the system | (100%) | 1 | |
| Acceptability | • Completeness of USI reporting forms<br>N: Number of filled entry<br>D: Total number of field entries on that form | 300/300<br>(100%) | 1 | 3 |
| | • Timeliness of USI data reporting<br>N: number of reports submitted timely<br>D: total number of reports | 97% of the reports were timely submitted | 1 | |
| | • Participation rate in the system<br>N: Number of interviewees accepting the system<br>D: Total number of interviewees who were evaluated via interview | 9/12(75%) | 1 | |
| | • Formal training given | No training is given to personnel working in the system | | |
| Timeliness | • The mean Time interval of taking action | • 1 day | 1 | 3 |
| | • The mean time interval between data collection and data sent to the next level and time data is entered in the system. | • 5 days | 1 | |
| Sensitivity | • The ability of RTK to test non-iodized salt | • RTK captures all non-iodized salt | 1 | 3 |

Key: N = Numerator

D = Denominator

In instances where the key findings did not support the indicator assessed for a particular attribute, that indicator received a score of zero (0). Attributes exhibiting an average score of more than two-thirds of the total score were considered major strengths of the system and received an overall score of 3. On the other hand, attributes with an average score of less than one-third were identified as major weaknesses, denoted by an overall score of 1. Attributes falling between these ranges, with scores between one-third and two-thirds, were categorized as relative strengths, signified by an overall score of 2.

These scores were assigned on a scale ranging from 1 to 3, reflecting performance levels from poorest to best, respectively [16].

### Ethical clearance

The study was conducted as part of the routine assessment activities of the TFNC at the lower administrative levels. It did not necessitate formal approval from an ethics review board. Instead, formal communication was established with regional and district authorities regarding the assessment of the surveillance system. Notably, all participants in this study were government officials. Furthermore, before conducting interviews, verbal consent was obtained from all participants. It is essential to know that the confidentiality of all participants was diligently maintained and assured.

## Results

This chapter serves the purpose of presenting the study's findings. It is structured into two main parts: **"Operation of the USI Surveillance System"** and **"Evaluation of System Attributes."**

The "Operation of the USI Surveillance System" details the actual functioning of the system as observed during the study. This observation of the system's real-world operation led to the identification of key findings and limitations.

On the other hand, the evaluation of "System Attributes" aimed to assess whether the surveillance system effectively fulfils its intended objectives.

### Operation of the USI surveillance system

The operation of the USI surveillance system encompasses several crucial elements. To begin, the system receives vital support in the form of funding, reagents, and testing equipment from government sources and development partners like Nutrition International (NI) and UNICEF.

One key aspect of the system involves the collection of data on salt iodization. These data are gathered from various sources, including schools, shops, and markets. Salt samples are collected and undergo on-site testing using a Rapid Test Kit (RTK). Subsequently, data on these samples are forwarded to the district level every month. At the district level, this data is aggregated and subsequently submitted to the regional and national levels on a quarterly basis. It's important to note that, despite the monthly data collection on salt iodization, there was no predefined target for the number of salt samples to be collected and tested at the ward and district levels. Such targets are essential for assessing performance indicators on a monthly and quarterly basis.

The responsibility for collecting data on salt iodization is shared between health officers and nutritionists. Health officers inspect and test salt sold in markets and shops, having the authority to take legal action against any violations, as stipulated by the Tanzania Food and Drug Authority (TFDA) Act section 105. On the other hand, nutritionists collect samples from school children, who serve as representatives of households. Nutritionists, however, lack

legal authority and mainly focus on testing for the presence of iodine in the salt and providing advice to enhance salt iodization coverage.

Additionally, TFNC plays a vital role by supplying essential tools at the regional level, such as RTKs and sealed plastic bags for transporting salt samples for content analysis. The distribution of these tools is managed from the regional level to the district level and further down to the ward level through Regional Nutrition Officers (RNuOs) and District Nutrition Officers (DNuOs), respectively.

Data from the district level is communicated to higher levels through IMES in DHIS2, with the data eventually being pooled by the Multisectoral Nutrition Information System (MNIS) at TFNC. Moreover, TFNC conducts quarterly supervision at the regional and district levels, while regional officials also conduct supervisory activities at the district and ward levels.

The USI surveillance system receives support from various stakeholders, both in terms of financial resources and structural assistance. These critical stakeholders include the Ministry of Health (MOH), Tanzania Food and Nutrition Centre (TFNC), the President's Office of Regional Administration and Local Government (PO-RALG), health officers, nutrition officers, Council Health Management Teams (CHMTs), and the Regional Health Management Teams (RHMTs).

## Key findings observed from the operation of the Universal Salt Iodization (USI) system

The key findings stemming from the operation of the Universal Salt Iodization (USI) system reveal critical areas of concern.

Firstly, the USI forms employed in data collection were found to be lacking in comprehensiveness. These forms contained a limited number of data variables, rendering them inadequate for meeting the system's requirements. The USI form consisted of just five entries, collecting data on the number of samples collected, the number of samples testing positive, the number of samples testing negative, the percentage of samples in each category, and the negatives. The significant shortcoming here is the omission of essential information that needs to be reported from lower levels to the national level. This missing information includes crucial data such as village and ward specifics (e.g., village names, the number of salt producers, the count of salt factories, etc.) and information regarding inventory and equipment status (for example, the availability and functionality of WYD machines and reagents).

Secondly, it was noted that the USI working definition did not align with certain country laws, particularly the Tanzania Food and Drug Authority (TFDA) Act of 2003. The USI working definition categorizes salt containing less than 15 ppm as unfit for human consumption. In contrast, the TFDA Act emphasizes packaging requirements (properly packed with a plastic liner to prevent iodine evaporation) and the proper storage of salt, which significantly impacts iodine preservation in salt. In Tanzania, the enforcement of salt iodization is primarily the responsibility of health officers and nutritionists. However, only health officers are authorized to enforce the Public Health Act (PHA of 2009) and the TFDA Act. This limitation, which prevents DNUOs from taking actions beyond testing for iodine presence in salt, poses a challenge to expanding iodized salt coverage due to the scarcity of health officers. It is imperative to extend the working definition to encompass matters related to salt storage and packaging, as these factors can also influence the iodine content in salt.

Thirdly, a misalignment was observed among data systems, including TFNC data system, the Multisectoral Nutrition Information System (MNIS), and the President's Office of Regional Administration and Local Government (PO-RALG) data system known as IMES.

Presently, TFNC and the Ministry of Health (MOH) are not receiving data from the PO-RALG system as initially intended (see Fig 2).

Lastly, a notable omission in the system was the failure to include urinary Iodine Excretion levels, which can have several consequences, primarily leading to an inadequate assessment of iodine status.

### Evaluation of system attributes

**Usefulness.** *Non-iodized salt detection.* Out of all the samples collected (1715) in 2021, the highest number of samples without iodine was found at Kasulu Town Council, followed by Kigoma Municipal and the lowest at Kibondo DC as shown in Fig 3.

### The usefulness of the Universal Salt Iodization (USI) surveillance system in attaining its objectives

**Usefulness.** The utility of the Universal Salt Iodization (USI) surveillance system in accomplishing its objectives was closely scrutinized. This system boasts five distinct objectives, which encompass providing clear guidance to USI stakeholders on the monitoring and evaluation of iodated salt at regional, council, and community levels, ultimately striving to virtually eliminate Iodine Deficiency Disorder (IDD). Additionally, these objectives aim to offer direction to USI stakeholders about their roles and responsibilities in monitoring and evaluation, as well as guide the collection of information of salt iodization. The system should also address the enhancement of the USI monitoring and evaluation system and describe the indicators employed in assessing IDD's magnitude at different stages of the USI program.

In this evaluation, it was discerned that, out of the five stated objectives, only one was effectively met during the assessed period. Specifically, the system proficiently reports the number of samples collected, both positive and negative, along with their respective percentages. However, the system fell short in terms of identifying the magnitude of the problem, as detailed in Table.

### Simplicity, flexibility, stability and representativeness

Respondents involved in the assessment indicated that the system was notably straightforward to operate. It required minimal time for analysis and report generation. Furthermore, the surveillance working definition was considered user-friendly. Nevertheless, a notable finding was that individuals working within the system did not receive any specialized training, as documented in Table 1. Regarding the issue of Flexibility, the system exhibited a commendable capacity to adapt to changes without encountering significant disruption. For instance, when the regulatory authority for salts transitioned from the Tanzania Food, Drugs, and Cosmetics Act (TFDA) to the Tanzania Bureau of Standards (TBS) in 2019, the system continued to operate seamlessly. Moreover, the system adheres to a standardized flow of information, and electronic data can be smoothly integrated into other systems, as outlined in Table 1. On the issue of Stability of the system it was revealed that data flow within the system adheres to a predefined set of standards. Data collection occurs on a routine monthly basis, extending from wards to district levels, and every quarter from the district level into the Integrated Management and Evaluation System (IMES). It's worth noting that the system relies on government funding to sustain its operations. In assessing its stability, respondents were queried regarding the system's operational continuity throughout the year. Impressively, all 12 interviewees affirmed that the system maintained performance levels exceeding 75%. Moreover, it was observed that there were no unscheduled outages or downtimes affecting the system's computers during the evaluation period, as detailed in Table 1. Moreover, it was found that the system

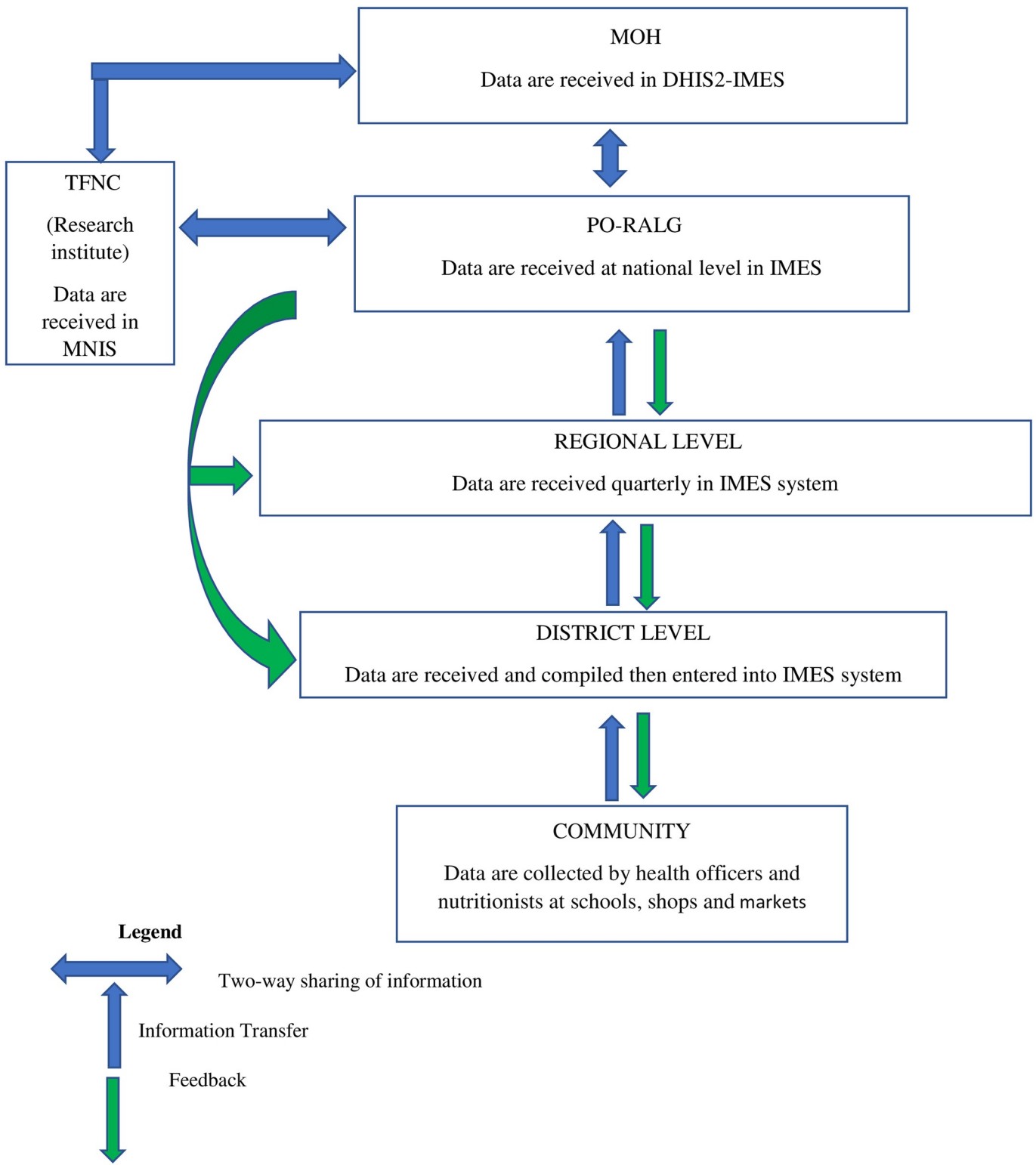

**Fig 2. USI Flow of information and feedback.**

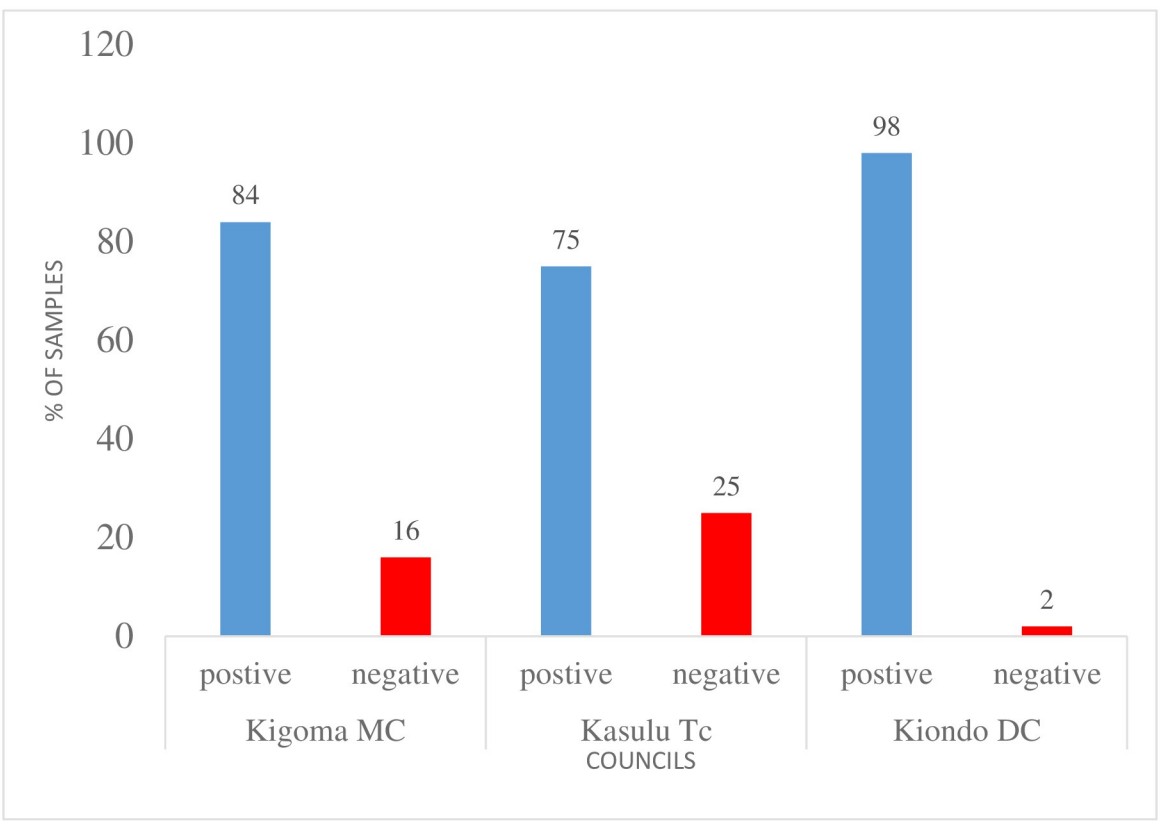

**Fig 3. Percentage of samples collected with positive and negative results at three districts in the Kigoma region from January-December 2021.**

effectively captures data on salts consumed by the community, i.e., the population under surveillance. The data documented in the system accurately represents the salts used by the community, as it is sourced from salt samples collected from schools, which serve as proxies for household salts, as well as samples procured from points of sale where household salt is purchased. However, an interesting observation was made—despite detecting various salts in the council, there was a lack of comprehensive descriptions regarding their origins or sources, as outlined in Table 1.

## Data quality, acceptability, sensitivity and timelines

All three evaluated indicators had good performance (completeness in entering the data in the system, Completeness in filing field entry and data accuracy), and the mean score performance was 100% (Table 2). Regarding the acceptability of the system it was observed that Out of the three indicators assessed, completeness, timeliness and participation rate, only participation rate had performance less than 100% (see Table 2). The overall acceptability was 91.6%. However, apart from Regional and District Nutrition Officers and a few health officers, it was observed that no other cadre/staff used the data or the system and all interviewees had never been given formal training in the Universal Salt Iodization (USI) surveillance system (Table 2). Regarding the Sensitivity of the system it was found that the system has been able to capture non-iodized salt every month during the time evaluated (Table 2). Lastly in regard of the timeliness of the system the following was found. Timeliness was evaluated by observing the time

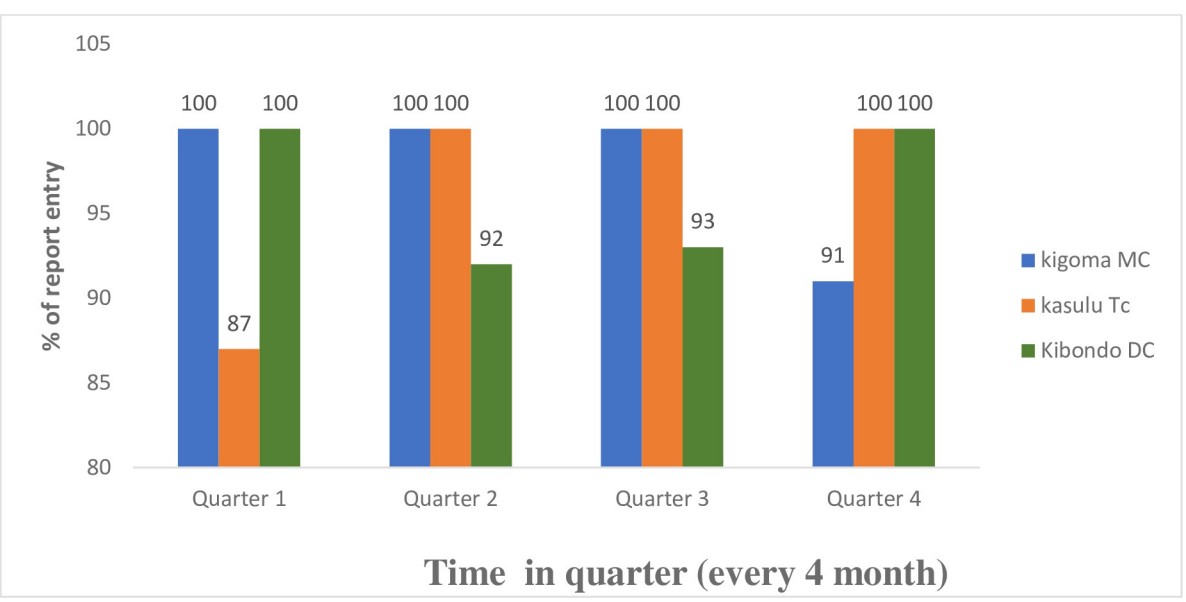

**Fig 4. Timeliness in data entry in three councils, Kigoma region 2021.**

interval between identifying a problem or issue and taking action, as well as timely reporting or submitting a report to the next level or in the system. The majority of the respondents were aware of the date they were supposed to send the reports to the next level. It was observed that reports from the ward level were submitted on time to the district level and that it took an average of 5 days for all of the reports to be entered into the system. Yet, several councils were spotted failing to submit reports into the system on time. The difference observed in different councils in different quarters was caused by ad hoc activities that some councils face at their level hence failing to enter the reports timely into the system. Moreover, the evaluation found that the actions are taken timely (immediately) on the same day they find the problem. The actions that were found to be taken include the provision of education to sellers and schoolchildren and the immediate removal of non-iodate salts from points of sale. (See Fig 4).

## Discussion

The USI surveillance system is the system that deals with the monitoring and evaluation of salt iodization in Tanzania. It deals with all salts for human consumption from the community level, shops, and production areas.

The utilization of non-iodized salt can have severe health repercussions, potentially leading to Iodine Deficiency Disorder (IDD). To combat this issue, it is crucial to ensure that iodized salt reaches a significant portion of households, aligning with the World Health Organization's (WHO) target of $\geq 90\%$ household coverage. Achieving this goal and preventing IDD hinges on the effectiveness of the Universal Salt Iodization (USI) surveillance system in any country including Tanzania.

In Tanzania, the USI surveillance system commenced in 2010, addressing concerns related to salt iodization by offering clear guidance on monitoring and evaluating iodated salt at various levels: regional, council, and community. The overarching objective is the virtual elimination of IDD.

A recent evaluation of the system revealed that a majority of its attributes (90%) performed well. However, the study also uncovered that the system fell short in meeting four out of five

objectives. This shortfall is primarily due to the absence of a comprehensive data collection tool and a lack of integration with other nutritional systems at the Ministry of Health and the Tanzania Food and Nutrition Centre (TFNC). This contrasts with findings from Congo [17] but somehow aligns with the study conducted in Ghana [16]. One of the strengths of the system is its simplicity, both in its working definition and in the collection and analysis of data, a quality also shared with the systems in Ghana and Congo [16, 17]. Furthermore, it is adaptable to changes and represents the salts used in the community, providing a degree of flexibility, as observed in the Tanzanian study [18]. Additionally, the system demonstrates stability as it relies on government funding and maintains a consistent flow of information, much like the Ghanaian system [16]. It is found to be acceptable for use by two types of users, nutritionists and health officers, though the legal mandate primarily rests with health officers.) [12, 17]. The system has good quality data with completeness and matching of hard copies and soft copies showing good performance. In line with the study conducted in Congo [17] but the result from this study was better compared with the study conducted in Ghana [16].

Despite these positive attributes, the system is hampered by its lack of a comprehensive data collection form. The current USI form contains only five entry variables, which inadequately capture important information necessary for reporting at various levels, such as village and ward details, and equipment status. This deficiency can negatively impact data quality and the system's usefulness. The five entry variables that collect information on number of samples collected, number of samples that tested positive and negative, percent of samples that tested positive and negative.

The USI surveillance form misses a lot of important information that is required to be reported from the lower level to the national level for action to be taken. Examples of important information missed are Village and Ward information (e.g., name of the village, number of salt producers' number of salt factories etc.), Knowing this information will help to identify where specifically the collected samples are coming from in the reporting facility. Other missing information is on inventories or equipment status (number of WYD machines if available/ working or not, reagents availability), this information is important to be collected because will give the actual situation of reagents and other USI tools so, those who are responsible for distribution may take action immediately. The system has a form that is not comprehensive and hinders it from providing comprehensive information to stakeholders for decision-making. Even if some of the attributes had a good performance the information generally collected by the system is not comprehensive to meet the need. Hence lacking a standard comprehensive form may affect data quality and acceptability of the results from the analysis of such quality and general usefulness of the system.

The working definition of USI also presents a challenge as it does not align with Tanzanian laws and regulations, e.g., the Tanzania Food and Drug Authority (TFDA) Act,2003 and the Salt Regulation Act,2010leading to discrepancies in mandates. Addressing this issue may require clarifying the roles of nutritionists and health officers or amending the working definition to align with legal requirements. In Tanzania, the issue concerning salt iodization is enforced by health officers and nutritionists, but the laws, only identify and give the mandate to health officers to deal with issues of salt iodization (TFDA Act section 105) [12–14], this being the case nutritionist cannot go beyond testing the presence of iodine in the salt and this is a challenge in improving iodized salt coverage. Moreover, the system working definition is missing key important parts that also affect the presence of iodine in the salt. The current working definition states that *salt containing <15 ppm is not fit for human consumption* but there are issues with storage, packaging and labeling that may affect salt content in the salt. Despite that, the majority of respondents claim that the working definition is simple to use but it lacks some very important parts like issues of proper storage, packing salts with plastic liner

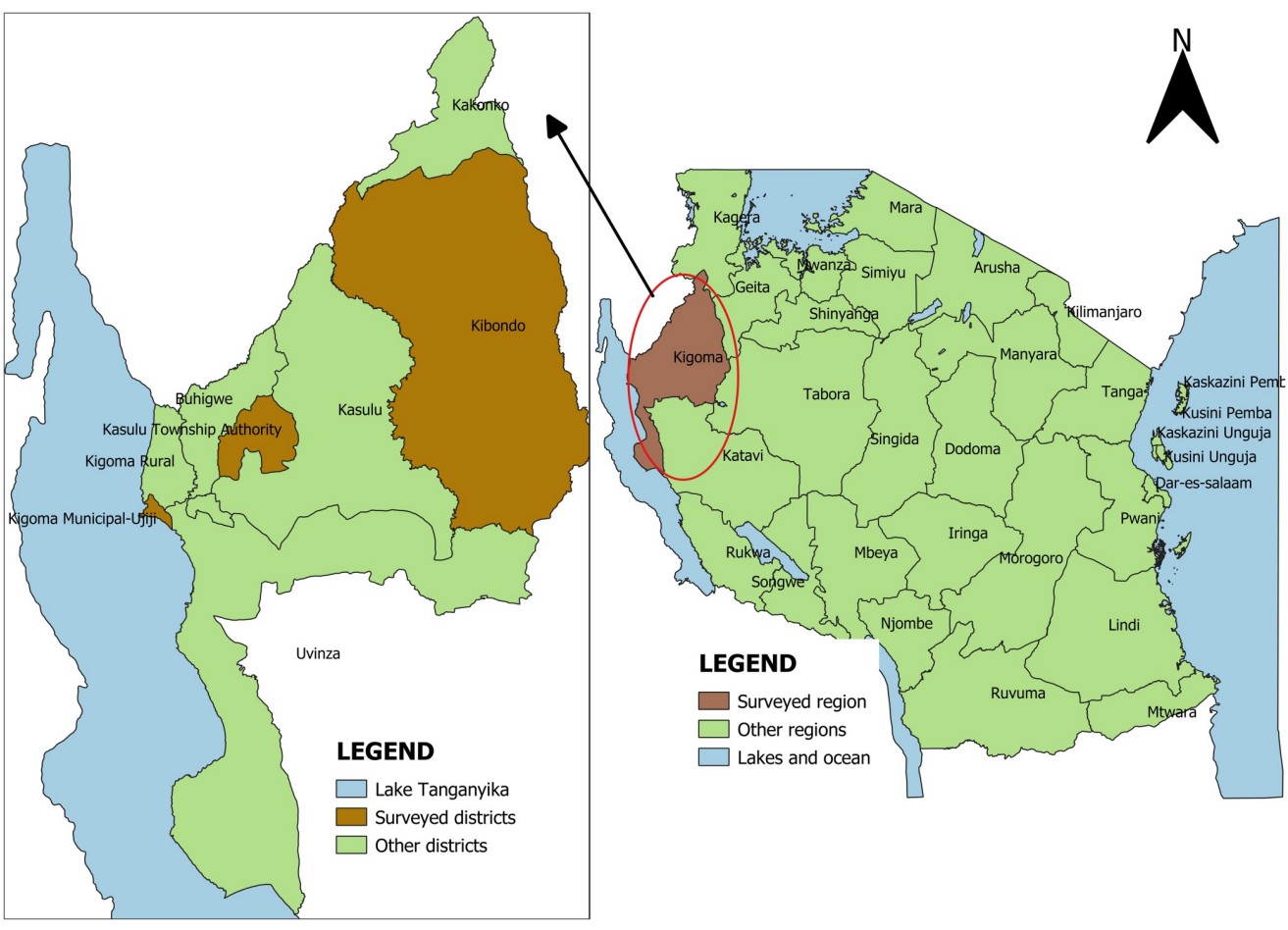

**Fig 5. A map of Tanzania showing the study area.**

and proper labeling should be added as required by TFDA Act of 2003 and Salt regulation. The definition is proposed to be *salts that are < 15 ppm, not well packed with a plastic liner to avoid iodine escape and improper storage of salt that will influence loss of iodine in the salt as not fit for human consumption*. In addition to that, to avoid confusion we, suggest that either Nutritionist should be identified by the law to work on salt iodization or should be exempted to leave the issues USI surveillance system to health officers only.

Additionally, the lack of data integration between the TFNC, Ministry of Health, and other relevant systems presents a roadblock to effective coordination and decision-making. The data systems of the Tanzania Food and Nutrition Centre, the Multisectoral Nutrition Information System, and the PO-RALG data system, the Integrated Management and Evaluation System (IMES) are not aligned. Presently, TFNC and the Ministry of Health (MOH) are not receiving data from the PO-RALG system, which was originally intended. This situation is consistent with the findings of a study conducted in Ghana (Fig 5).

Efforts are ongoing to facilitate the integration of these systems, but the misalignment extends to the structure of the data collection forms used by these entities. TFNC employs a comprehensive data collection tool that captures almost all the necessary information required by the system. In contrast, PO-RALG, responsible for gathering information from lower administrative levels (district and regional), utilizes a form with only 15 entry fields. At the

district level, it was observed that the tool developed by PO-RALG was being used due to their direct involvement with the system (as depicted in Fig 5). However, this form falls short in capturing as much information as the one developed by TFNC. This misalignment creates difficulties for TFNC, MOH, and other stakeholders in obtaining the necessary data.

To address this issue, it is imperative to expedite the process of integrating IMES with other systems, including MNIS from TFNC and any other systems that rely on USI data. The study has clearly demonstrated that these systems are not effectively communicating with one another. Consequently, TFNC and the Ministry of Health are not receiving the essential information pertaining to salt iodization. Accumulating information without disseminating it to stakeholder's hampers efforts to improve iodized salt coverage in the country.

Lastly, the absence of clear targets for collecting and testing salt samples makes assessing the system's performance at the council level difficult. Setting such targets in collaboration with TFNC is essential to better monitor and improve salt iodization coverage.

## Concluding remarks

Although the system attributes were found to be working overall well, for proper surveillance of USI system, the core attribute needs to be strengthened, key variable that measures the system performance must be included from the primary data source and well-integrated within primary source to MoH and TFNC to be able to attain 90% global coverage. Strengthening USI surveillance is key for the country to attain a universal coverage target of 90%.

Additionally, the study faces some limitation. First, some of the health officers and nutritionists were unable to enter the system, forcing us to conduct physical interviews, which took more time. Second, the evaluation focused on document review and use of RTK only, the system does not include biomarker assessment nor other laboratory test.

Despite the limitations encountered during the study, it also exhibited strengths that enhance its utility. First, this study was the first USI surveillance system evaluation; hence the results of this study will trigger further study in this area. Second, the results of this evaluation will enable USI program to improve the system.

## Recommendations

Following our evaluations, we recommend the following to be done to make sure USI surveillance system work properly and achieve its intended objectives;

## At national level

- TFNC and MoH should review the standard working definition of iodated salt to include considerations for storage, labeling, and packaging. Additionally, ensure that health officers are trained and oriented on the use of this new standard working definition of iodated salt.

- TFNC, MoH, and PO-LARG to collaboratively create a single, comprehensive reporting form for monitoring and reporting on universal salt iodization (USI) activities. This will help in improving the uniformity and correctness of the data reported, reducing the stress on lower-level reporting.

- PO-LARG to establish specific targets for each council related to the quantity of iodated salt to be produced or distributed. These targets will aid in monitoring and measuring performance at the council level.

- TFNC and PO-LARG to ensure availability of salt testing equipment and reagents, including rapid tests, to health management teams at the lower levels. This provision will enhance the

capacity for timely testing and collection of quality data on the status of salt consumed by communities.

### Regional and council (district) level

- Region Health Management Teams (RHMT) and Council Health Management Teams (CHMT) must shift from relying solely on integrated supervision for USI monitoring and instead focus on USI-related variables. Consider conducting dedicated USI monitoring at least twice a year, separate from other integrated checklists.

## Supporting information

**S1 Dataset.**
(XLSX)

## Acknowledgments

We would want to thank everyone who took the time to ensure that this task was completed. Special appreciation to Dr. Germana Leyna, TFNC's managing director, all TFNC staff members, RHO and RNuO in Kigoma, and all Kigoma health officers and nutritionists. Finally, we thank the TFELTP and the CDC for close supervision of this evaluation. We applaud your efforts and pray that the Lord blesses you all.

## Author Contributions

**Conceptualization:** David Mahwera, Erick Killel, Ninael Jonas, Adam Hancy, Anna Zangira, Aika Lekey, Rose Msaki, Doris Katana, Rogath Kishimba, Debora Charwe, Fatma Abdallah, Geofrey Chiduo, Ray Masumo, Germana Leyna, Geofrey Mchau.

**Data curation:** David Mahwera, Erick Killel, Ninael Jonas, Adam Hancy, Rose Msaki, Debora Charwe, Geofrey Chiduo, Germana Leyna, Geofrey Mchau.

**Formal analysis:** David Mahwera, Erick Killel, Ninael Jonas, Adam Hancy, Anna Zangira, Aika Lekey, Rose Msaki.

**Investigation:** David Mahwera, Erick Killel, Ninael Jonas, Adam Hancy, Aika Lekey, Rose Msaki, Doris Katana, Debora Charwe, Fatma Abdallah, Ray Masumo, Germana Leyna, Geofrey Mchau.

**Methodology:** David Mahwera, Erick Killel, Ninael Jonas, Adam Hancy, Anna Zangira, Aika Lekey, Rose Msaki, Doris Katana, Rogath Kishimba, Debora Charwe, Fatma Abdallah, Ray Masumo, Germana Leyna, Geofrey Mchau.

**Project administration:** David Mahwera, Anna Zangira, Aika Lekey, Doris Katana, Debora Charwe, Geofrey Chiduo, Ray Masumo, Geofrey Mchau.

**Resources:** David Mahwera, Erick Killel, Ninael Jonas, Anna Zangira, Aika Lekey, Rose Msaki, Doris Katana, Rogath Kishimba, Debora Charwe, Geofrey Chiduo.

**Software:** David Mahwera, Ninael Jonas, Adam Hancy.

**Supervision:** Rose Msaki, Debora Charwe, Geofrey Chiduo, Ray Masumo, Germana Leyna, Geofrey Mchau.

**Validation:** David Mahwera, Erick Killel, Adam Hancy, Rose Msaki, Doris Katana, Debora Charwe, Geofrey Chiduo, Germana Leyna, Geofrey Mchau.

**Visualization:** David Mahwera, Adam Hancy, Rose Msaki, Doris Katana, Geofrey Chiduo.

**Writing – original draft:** David Mahwera, Erick Killel, Ninael Jonas.

**Writing – review & editing:** Rose Msaki, Doris Katana, Rogath Kishimba, Debora Charwe, Fatma Abdallah, Geofrey Chiduo, Ray Masumo, Germana Leyna, Geofrey Mchau.

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
