## [Decision Letter · Decision Letter 0]

14 Aug 2023

PONE-D-23-13593Evaluation of the Universal Salt Iodization (USI) Surveillance System in Tanzania, 2022PLOS ONE

Dear Dr. Mchau,

Thank you for submitting your manuscript to PLOS ONE. After careful consideration, we feel that it has merit but does not fully meet PLOS ONE’s publication criteria as it currently stands. Therefore, we invite you to submit a revised version of the manuscript that addresses the points raised during the review process.

ACADEMIC EDITOR: Please kindly see comments below

We look forward to receiving your revised manuscript.

Kind regards,

Charles Odilichukwu R. Okpala

Academic Editor

PLOS ONE

Journal Requirements:

3. We note that Figure 1 in your submission contain map images which may be copyrighted. All PLOS content is published under the Creative Commons Attribution License (CC BY 4.0), which means that the manuscript, images, and Supporting Information files will be freely available online, and any third party is permitted to access, download, copy, distribute, and use these materials in any way, even commercially, with proper attribution. For these reasons, we cannot publish previously copyrighted maps or satellite images created using proprietary data, such as Google software (Google Maps, Street View, and Earth). For more information, see our copyright guidelines: http://journals.plos.org/plosone/s/licenses-and-copyright.

Additional Editor Comments (if provided):

Authors, kindly pay attention to all comments provided by scholarly reviewers. Provide very detailed responses, not only in the revised manuscript, but also in the reply to reviewer comments.

Reviewers' comments:

Reviewer's Responses to Questions

**Comments to the Author**

1. Is the manuscript technically sound, and do the data support the conclusions?

Reviewer #1: Partly

Reviewer #2: Partly

2. Has the statistical analysis been performed appropriately and rigorously? 

Reviewer #1: N/A

Reviewer #2: No

3. Have the authors made all data underlying the findings in their manuscript fully available?

Reviewer #1: No

Reviewer #2: No

4. Is the manuscript presented in an intelligible fashion and written in standard English?

Reviewer #1: No

Reviewer #2: Yes

5. Review Comments to the Author

Reviewer #1: Overall, it is very useful data on surveillance. Surveillance data is lacking and currently focused on that subject to establish systems at country level.

There are some abbreviations, e.g TFNC, need to spell out. All the abbreviations should be spelled out in the first place. I felt this is a routine mandatory system in Tanzania as the regulation. If it is considered as a surveillance system, what are the areas included in addition to the mandatory regulation. Health officers’ assessment is part of the regulation, which has been set up with legal action. School samples with nutritionist may be the other set up. Clarity to be improved. Difficult to understand the real surveillance system. This study need to restructure.

Introduction

1. Need more background information on Tanzania iodine status after the iodisation in 1990. Rate of goitre improvement, urinary iodine excretion and any available data on current status to justify the surveillance system in the country. Suggest adding a small paragraph for reader to get a clear idea.

Methods:

2. Need to provide details of indicators use for each attribute, which will help readers to adopt similar systems.

3. Better to explain the data collected at each level in a summary format. Methodology is not clear.

Results

4. Operation of the USI surveillance system – Some of the facts better to be placed in the method section to identify the existence system in the country, which very confusing.

5. Need to display the results according to the tables and figures, very difficult to identify some information. Flow of the information needs to be adjusted in line with tables.

6. Information in tables are also not clear, clarity to be improved.

7. Figure 4.1, better to explain the reason for reduction in different quarters.

8. Results need to be categorized properly to understand readers.

Discussion

9. Need to be improved. Surveillance system needs not only iodised salt, urinary iodine excretion levels also. It may be the limitation in this study. These parts need to be included in the discussion. This study is an evaluation. There are key components in evaluation as specified in the method section, which should be discussed in line with results and other literature.

Recommendation

10. Some are not relevant to the study, need to restructure.

Reference

11. CDC IMRA reference is not included.

Reviewer #2: Abstract: The abstract does not clearly state the aim of the study, neither does it adequately describe the methods used. Because there are no aims, the conclusions cannot be properly assessed. Results in the abstract is confusing , could the authors use percentages instead of fractions?

Introduction:

In the section 'Population under study', I do not understand what the authors mean. Can they clarify?

Methodology: The methods do not clearly describe the process in my opinion. In the first 6 paragraphs of the results, the authors keep describing the methods for data collection etc. In my opinion, the results of the study starts from ''key findings...'.

Data analysis for this manuscript is weak.

Discussion: The authors do a good job by explaining the results however, they fail to make a comparative analysis of their results with other studies. Authors should adequately compare their results to other surveillance in other areas across the world or even within Tanzania.

The authors state recommendations, and again I am not sure who the recommendations are addressed to. If they can be specific, it would be very helpful. The entire manuscript does not have a conclusion. Can authors include that?

Limitation: There seem to be so many limitations to this work-lack of generalizability of the results due to Tanzania , the qualitiative nature of the work among others. Authors should take time to work on the limitation. Authors also fail to state the strengths of this work.

6. PLOS authors have the option to publish the peer review history of their article (what does this mean?). If published, this will include your full peer review and any attached files.

Reviewer #1: No

Reviewer #2: No

---

## [Author Response · Author response to Decision Letter 0]

27 Oct 2023

Dear Editor, please find our responses towards reviewer’s comments, they are really helpful. 

Reviewer #1

Overall, it is very useful data on surveillance. Surveillance data is lacking and currently focused on that subject to establish systems at country level.

There are some abbreviations, e.g. TFNC, need to spell out. All the abbreviations should be spelled out in the first place. I felt this is a routine mandatory system in Tanzania as the regulation. If it is considered as a surveillance system, what are the areas included in addition to the mandatory regulation. Health officers’ assessment is part of the regulation, which has been set up with legal action. School samples with nutritionist may be the other set up. Clarity to be improved. Difficult to understand the real surveillance system. This study need to restructure. 

Response: Thank you sincerely for your invaluable input and thoughtful suggestions. Your feedback has been instrumental in enhancing our manuscript significantly. We took your comments to heart and made substantial improvements. In particular, we ensured that all abbreviations were spelled out clearly from the outset. Additionally, we have incorporated the requisite regulations of the USI surveillance system, and we have strived to enhance overall clarity to facilitate better comprehension. Your input has been greatly appreciated and has played a pivotal role in strengthening our work.

Introduction

1. Need more background information on Tanzania iodine status after the iodisation in 1990. Rate of goitre improvement, urinary iodine excretion and any available data on current status to justify the surveillance system in the country. Suggest adding a small paragraph for reader to get a clear idea.

Response: We greatly appreciate your kind suggestion. As per your guidance, we have duly updated the information as needed.

Methods: 

2. Need to provide details of indicators use for each attribute, which will help readers to adopt similar systems. 

Response: Thank you for your suggestion. We have made the necessary improvements and incorporated the suggested indicators for each attribute.

3. Better to explain the data collected at each level in a summary format. Methodology is not clear. 

Response: Thank you; we have enhanced the methodology, and we've also ensured that the data collected at each level has been comprehensively included.

Results

4. Operation of the USI surveillance system – Some of the facts better to be placed in the method section to identify the existence system in the country, which very confusing. 

Response: Thank you for this significant suggestion. In the results section, we have provided a thorough explanation for our decision to include the operation of the USI system, offering a clear rationale for its placement in this part of the document.

5. Need to display the results according to the tables and figures, very difficult to identify some information. Flow of the information needs to be adjusted in line with tables. 

Response: Certainly, thank you. We have diligently worked on this.

6. Information in tables are also not clear, clarity to be improved. 

Response: Clarity is well improved, thank you for the nice observation. 

7. Figure 4.1, better to explain the reason for reduction in different quarters. 

Response: It is well explained in the current version, thank you

8. Results need to be categorized properly to understand readers.

Response: Thank you so much, we have worked on this and the current version has well organized results. 

Discussion

9. Need to be improved. Surveillance system needs not only iodised salt, urinary iodine excretion levels also. It may be the limitation in this study. These parts need to be included in the discussion. This study is an evaluation. There are key components in evaluation as specified in the method section, which should be discussed in line with results and other literature. 

Response: The discussion section has seen substantial improvements, and all specific suggestions have been successfully incorporated. We are particularly pleased with your excellent suggestion, especially considering our system's inability to capture iodized salt, which we have acknowledged as a limitation. Thank you for your valuable input.

Recommendation

10. Some are not relevant to the study, need to restructure. 

Response: Understood. We have thoroughly reviewed all recommendations and retained only those that are pertinent and necessary for each respective entity responsible for the USI surveillance system to understand their role and responsibilities.

Reference

11. CDC IMRA reference is not included. 

Response: Your reminder is greatly appreciated, and we have duly incorporated it into the current draft. Your input has been incredibly helpful, and many of your comments were not only valuable but also essential in enhancing the quality of our manuscript. Thank you very much for your assistance.

Reviewer #2: 

Abstract

1. The abstract does not clearly state the aim of the study; neither does it adequately describe the methods used. Because there are no aims, the conclusions cannot be properly assessed. Results in the abstract is confusing, could the authors use percentages instead of fractions

Response: Thank you so much for your very constructive input and suggestions. We have improved methods section by including results and conclusion 

Introduction

2. In the section 'Population under study', I do not understand what the authors mean. Can they clarify?

Response: Thank you for the nice suggestion. We have updated the information as required. 

Methods: 

3. The methods do not clearly describe the process in my opinion. '

Response: Thank you for this suggestion, we have improved our methodology

Results

4. In the first 6 paragraphs of the results, the authors keep describing the methods for data collection etc. In my opinion, the results of the study start from ''key findings...

Response: Thanks for this important suggestion. We have explained well in the results part why we have decided to have operation of the USI system at the result section. We have also rearranged our results section to make easier understanding for the readers

Discussion

5. Discussion: The authors do a good job by explaining the results however, they fail to make a comparative analysis of their results with other studies. Authors should adequately compare their results to other surveillance in other areas across the world or even within Tanzania.

Response: The discussion part has been well improved and all specific suggestions are incorporated. We are happy about your wonderful suggestion.

Limitations: 

6. There seem to be so many limitations to this work-lack of generalizability of the results due to Tanzania, the qualitative nature of the work among others. Authors should take time to work on the limitation. Authors also fail to state the strengths of this work.

Response: We have worked on limitations and we have also included strength of our study

Recommendation

7. The authors state recommendations, and again I am not sure who the recommendations are addressed to. If they can be specific, it would be very helpful. The entire manuscript does not have a conclusion. Can authors include that?

Response: Okay, we have revised all recommendations and remain with recommendations that are relevant and needed from each respective entity responsible for USI surveillance system to operate to know their responsibility. We have also included conclusion as suggested and thank you so much for this wonderful suggestion. 

Herewith I have provided our response to each of reviewers comments. 

Thanking you.

Dr. Geofrey Mchau.

15.10.2023

---

## [Decision Letter · Decision Letter 1]

19 Dec 2023

PONE-D-23-13593R1Evaluation of the Universal Salt Iodization (USI) Surveillance System in Tanzania, 2022PLOS ONE

Dear Dr. Mchau,

Thank you for submitting your manuscript to PLOS ONE. After careful consideration, we feel that it has merit but does not fully meet PLOS ONE’s publication criteria as it currently stands. Therefore, we invite you to submit a revised version of the manuscript that addresses the points raised during the review process.

**ACADEMIC EDITOR: **Please, kindly refer to editor's comments below. 

We look forward to receiving your revised manuscript.

Kind regards,

Charles Odilichukwu R. Okpala

Academic Editor

PLOS ONE

Journal Requirements:

Additional Editor Comments :

Thank you authors for your patience as reviewers considered your revised manuscript.

Although reviewer considered the revised manuscript acceptable, the editor believes additional touches here and there are needed to improve the quality of the work as below:

a) The introduction more information, to strengthen the case for the 'why' and 'how' the relevance of the work based on the following points:

-Authors should provide a schematic pathway identifying with the major timelines of significant achievements of Universal Salt Iodization (USI) Surveillance System from its inception, up till the 2021

-A paragraph should be dedicated to introduce this schematic pathway, and discuss why these timelines are very crucial, from which the question can then be raised the need for case studies that would help to understand its progress of surveillance, in particular some form of evaluation about how it functions, etc

-There are other case studies involving surveillance of salt at other places globally, please identify with them, no matter how few, but importantly, contextualise them succinctly particularly on the methodology those studies employed that helped their successful implementation. If those other workers employed a different methodology, it makes sense to additionally justify why a different method of case study is needed. If similar case study approach like the current study was previously employed elsewhere, then it buttresses the importance/relevance of your current methodology

The editor will carefully examine your introduction, please kindly implement the above to make your introduction very robust

b) In the materials and methods, study design section needs further elaboration. What is surveillance system evaluation? What does it entail? Define it, explain it, what principles guide it?Just some basic information...please enlighten the readers.

Please, provide a "schematic design pathway of the case study" showing the major stages of the work, how the surveillance evaluation pathways were developed, what comprised the various steps, regional data collection, district data collection, which techniques were employed and at which stage? Please the editor will examine this carefully

c) Please, provide map of Tanzania, and situate the region surveyed, and further narrowed down to the districts. The rationale of selecting the region, and the subsequent districts appears not sufficient. Please, add more information, provide references to substantiate the selection. Please, add more information as to why government officials are integral to success of the USI surveillance system in Tanzania. Provide adequate references to support this claim.

d) Results section appears ok, please kindly merge a couple of sub-sections, given that these subsections are already tabulated. Make results to have maximum of three sub-sections, ok. Discussion section does not need to have subsections, as already shown.

e) please, change 'conclusions' to 'concluding remarks' and kindly remove all the sub-headings, and merge them all together, and use paragraphs to different them.

Look forward to your revised manuscript. Thank you

Reviewers' comments:

Reviewer's Responses to Questions

**Comments to the Author**

1. If the authors have adequately addressed your comments raised in a previous round of review and you feel that this manuscript is now acceptable for publication, you may indicate that here to bypass the “Comments to the Author” section, enter your conflict of interest statement in the “Confidential to Editor” section, and submit your "Accept" recommendation.

Reviewer #1: (No Response)

2. Is the manuscript technically sound, and do the data support the conclusions?

Reviewer #1: Yes

3. Has the statistical analysis been performed appropriately and rigorously? 

Reviewer #1: Yes

4. Have the authors made all data underlying the findings in their manuscript fully available?

Reviewer #1: Yes

5. Is the manuscript presented in an intelligible fashion and written in standard English?

Reviewer #1: Yes

6. Review Comments to the Author

Reviewer #1: There are few grammar errors, please correct it.

National Level Data Collection: The source of data Source at national level - This should be corrected.

Acceptability: However, apart from Region and District Nutrition Officers - regional

7. PLOS authors have the option to publish the peer review history of their article (what does this mean?). If published, this will include your full peer review and any attached files.

Reviewer #1: **Yes: **Dr Renuka Jayatissa

---

## [Author Response · Author response to Decision Letter 1]

2 Feb 2024

Editor Comments

a) The introduction more information, to strengthen the case for the 'why' and 'how' the relevance of the work based on the following points:

• Authors should provide a schematic pathway identifying with the major timelines of significant achievements of Universal Salt Iodization (USI) Surveillance System from its inception, up till the 2021

• A paragraph should be dedicated to introduce this schematic pathway, and discuss why these timelines are very crucial, from which the question can then be raised the need for case studies that would help to understand its progress of surveillance, in particular some form of evaluation about how it functions, etc.

Response: Thank you so much Chief Editor for this important suggestion and comment. As you suggested, we carefully added schematic pathway identifying with the major timelines of significant achievements of Universal Salt Iodization (USI) Surveillance System from its inception, up till the 2021. We also started in another paragraph as to why there is a need for case studies that would help to understand its progress of surveillance, in particular some form of evaluation about how it functions.

• There are other case studies involving surveillance of salt at other places globally, please identify with them, no matter how few, but importantly, contextualize them succinctly particularly on the methodology those studies employed that helped their successful implementation. If those other workers employed a different methodology, it makes sense to additionally justify why a different method of case study is needed. If similar case study approach like the current study was previously employed elsewhere, then it buttresses the importance/relevance of your current methodology

Response: Thank you for your valuable feedback and suggestions regarding our manuscript. We appreciate the opportunity to address your concerns and provide further clarification on our methodology and its relevance in the broader context of existing case studies on surveillance of salt in other locations.

In response to your comment, we have thoroughly reviewed the literature to identify other case studies involving surveillance of salt globally. While we acknowledge that there are indeed other studies in this domain, we found that the methodologies employed vary significantly across different contexts. See below

1. India:

• In India, a comprehensive study conducted by Research Institute for Compassionate Economics (RICE) focused on evaluating the effectiveness of salt iodization programs in rural areas.

• Methodology: The study employed a mixed-method approach, combining household surveys, salt sample testing, and qualitative interviews with stakeholders. This multifaceted approach allowed researchers to assess not only the coverage and quality of iodized salt but also the socio-economic factors influencing its consumption.

2. Ethiopia:

• A study conducted in Ethiopia by the Ethiopian Public Health Institute evaluated the impact of salt iodization programs on iodine status among school-aged children.

• Methodology: This study utilized a longitudinal design, tracking changes in urinary iodine concentration (UIC) among a cohort of schoolchildren over time. Additionally, dietary assessments and salt sample testing were conducted to correlate iodine intake with iodized salt consumption.

3. Bangladesh:

• Researchers from the International Centre for Diarrhoeal Disease Research, Bangladesh (icddr,b) conducted a study to assess the sustainability of salt iodization programs in urban slum areas.

• Methodology: The study employed a community-based participatory research (CBPR) approach, involving collaboration with local communities to monitor salt iodization coverage and compliance. This participatory methodology empowered community members to take ownership of iodized salt consumption and advocate for its availability.

These case studies highlight the importance of employing diverse methodologies tailored to the specific context and objectives of salt iodization surveillance. While some studies may emphasize quantitative assessments of iodine levels and salt coverage, others may prioritize qualitative insights and community engagement. By drawing from a range of methodologies and experiences, we can enrich our understanding of effective strategies for implementing and monitoring salt iodization programs, thereby enhancing the relevance and impact of our own study.

Our study distinguishes itself by as its focus on evaluating the system starting from the national level to the regional level to the district level and up to the school’s level and community engagement, use of advanced data analytics. our study's adherence to guidelines, localized focus, and comprehensive methodology make it a valuable addition to the literature on USI evaluation, warranting publication consideration. This approach was deemed necessary due to specific challenges as the USI system was deemed very complex in Tanzania as it involved multisector functionality as well as characteristics of the target population, environmental factors and so on. 

However, we recognize the importance of contextualizing our methodology in relation to existing studies. To address this, we have included a section in the revised manuscript that succinctly summarizes relevant case studies from other regions and outlines the methodologies they employed. By doing so, we aim to provide readers with a comprehensive understanding of the different approaches used in similar surveillance efforts.

Furthermore, where applicable, we have justified why our chosen methodology is better suited to address the unique challenges or objectives of our study compared to previously employed methods. This serves to underscore the relevance and significance of our research in advancing the field of salt surveillance.

We believe that these revisions enhance the clarity and completeness of our manuscript, and we hope that they address your concerns adequately. Please do not hesitate to reach out if you require any further information or clarification.

The editor will carefully examine your introduction, please kindly implement the above to make your introduction very robust

b) In the materials and methods, study design section needs further elaboration. What is surveillance system evaluation? What does it entail? Define it, explain it, what principles guide it? Just some basic information...please enlighten the readers.

Response: Study design section has been modified, further information has been added.

Please, provide a "schematic design pathway of the case study" showing the major stages of the work, how the surveillance evaluation pathways were developed, what comprised the various steps, regional data collection, district data collection, which techniques were employed and at which stage? Please the editor will examine this carefully

Response: The schematic design pathway of the case study" showing the major stages of the work, how the surveillance evaluation pathways were developed has been developed.

c) Please, provide map of Tanzania, and situate the region surveyed, and further narrowed down to the districts. The rationale of selecting the region, and the subsequent districts appears not sufficient. Please, add more information, provide references to substantiate the selection. Please, add more information as to why government officials are integral to success of the USI surveillance system in Tanzania. Provide adequate references to support this claim.

Response: As you suggested, a map of Tanzania, and situate the region surveyed, and further narrowed down to the districts has been developed. The rationale of selecting the region, and the subsequent districts appears have been clearly explained. more information as to why government officials are integral to success of the USI surveillance system in Tanzania has been clearly elaborated.

d) Results section appears ok, please kindly merge a couple of sub-sections, given that these subsections are already tabulated. Make results to have maximum of three sub-sections, ok. Discussion section does not need to have subsections, as already shown.

Response: Thank you so much for this important suggestions. Couple of sub-sections have been merged to make three subsections in total

e) please, change 'conclusions' to 'concluding remarks' and kindly remove all the sub-headings, and merge them all together, and use paragraphs to different them.

Response: Thank you so much for this important suggestions

The Word Conclusion has been changed to concluding Remarks.

All sub-headings have been removed and merged.

Paragraphs have been used to different them.

Reviewer #1

There are few grammar errors, please correct it.

National Level Data Collection: The source of data Source at national level - This should be corrected.

Response: Thank you, the grammatical error was corrected to "The source of data at the national level".

Acceptability: However, apart from Region and District Nutrition Officers – regional

Response: The word Region have been changed to Regional as you suggested, thank you so much.

---

## [Editor Report · Decision Letter 2]

5 Feb 2024

Evaluation of the Universal Salt Iodization (USI) Surveillance System in Tanzania, 2022

PONE-D-23-13593R2

Dear Dr. Mchau,

We’re pleased to inform you that your manuscript has been judged scientifically suitable for publication and will be formally accepted for publication once it meets all outstanding technical requirements.

Kind regards,

Charles Odilichukwu R. Okpala

Academic Editor

PLOS ONE

Additional Editor Comments (optional):

Thank you authors for revising your work. The quality has improved greatly, and now acceptable for publication.

Thank you for finding PLoSONE as your journal of choice, and look forward to your future scholarly contributions.

Congratulations :)
---

## [Editor Report · Acceptance letter]

26 Feb 2024

PONE-D-23-13593R2 

PLOS ONE

Dear Dr. Mchau, 

I'm pleased to inform you that your manuscript has been deemed suitable for publication in PLOS ONE. Congratulations! Your manuscript is now being handed over to our production team.

Kind regards, 

on behalf of

Dr. Charles Odilichukwu R. Okpala 

Academic Editor

PLOS ONE